# KRAS Copy Number Gain in Cell-Free DNA Analysis-Based Liquid Biopsy of Plasma and Bile in Patients with Various Pancreatic Neoplasms

**DOI:** 10.3390/ijms26188763

**Published:** 2025-09-09

**Authors:** Mark Jain, David Atayan, Tagir Rakhmatullin, Tatiana Dakhtler, Victoria Inokenteva, Pavel Popov, Aleksandr Farmanov, Mikhail Viborniy, Iuliia Gontareva, Larisa Samokhodskaya, Vyacheslav Egorov

**Affiliations:** 1University Clinic, Lomonosov Moscow State University, 119992 Moscow, Russia; tagir.rakhmatullin@internet.ru (T.R.); inokentyeva_victoria@bk.ru (V.I.); slm@fbm.msu.ru (L.S.); 2Joint Stock Company “Ilyinsky Hospital”, 143421 Moscow, Russia; d.atayan@ihospital.ru (D.A.); t.dakhtler@ihospital.ru (T.D.); popovpavel@mail.ru (P.P.); a.farmanov@ihospital.ru (A.F.); m.vyborniy@ihospital.ru (M.V.); y.gontareva@ihospital.ru (I.G.); v.egorov@ihospital.ru (V.E.)

**Keywords:** liquid biopsy, *KRAS*, copy number gain, copy number variation, cell-free DNA, bile, pancreatic cancer, pancreatic ductal adenocarcinoma, intraductal papillary–mucinous neoplasm

## Abstract

Cell-free DNA (cfDNA) analysis-based liquid biopsy is a rapidly emerging diagnostic and prognostic tool in pancreatic ductal adenocarcinoma (PDAC). *KRAS* point mutations are the main biomarkers used for the detection of tumor cfDNA. However, there is another less studied yet frequent genetic alteration in this gene, namely copy number gain (CNG). The aim of this study was to evaluate the diagnostic and prognostic potential of *KRAS* CNG analysis in plasma and bile of patients with PDAC using ddPCR. This study included healthy volunteers (*n* = 69), patients with PDAC (*n* = 94), and other pancreatic neoplasms (OPN) (*n* = 17). The sensitivity and specificity of *KRAS* CNG compared to the control group were 16% and 100% (AUC-ROC—0.580), and compared to the OPN group, 16% and 94% (AUC-ROC—0.554), respectively. Addition of KRAS point mutations to the analysis increased the sensitivity to 65% (AUC-ROC—0.824 and 0.801, respectively). Bile exhibited an equal *KRAS* CNG detection rate compared to plasma (20% vs. 16%). *KRAS* CNG was not associated with clinical parameters, except prognosis. The probability of survival was worse in patients with *KRAS* CNG (HR—3.54; 95% CI: 1.55–8.12; *p* = 0.001). *KRAS* CNG in cfDNA might be a promising biomarker for both diagnostic and prognostic purposes in PDAC.

## 1. Introduction

According to the Global Cancer Statistics (GLOBOCAN) 2022, pancreatic cancer (PC) ranks 12th in incidence and 6th in mortality [1]. This disease is responsible for approximately 5% of all cancer-related deaths worldwide, and it is estimated that its share will continue to grow over time [2]. The overall 5-year survival rate in PC is about 10% (ranging from 2% to 15%) [3,4]. Such a poor prognosis for this cancer is largely due to its late diagnosis: the overwhelming majority of patients (more than 80%) present with a locally advanced or metastatic disease at the time of the initial diagnosis [5]. Unfortunately, at early stages PC lacks specific symptoms, and efficient screening programs are not yet developed.

PC is a heterogeneous disease that includes tumors of various origins and localizations. The most common neoplasm of this organ is the pancreatic ductal adenocarcinoma (PDAC), which accounts for more than 90% of all PC cases [6]. Other less frequent neoplasms of the pancreas include various malignant tumors (such as squamous cell carcinoma, colloid carcinoma, neuroendocrine cancer, etc.) and benign tumors (such as intraductal papillary–mucinous neoplasm (IPMN), serous cystadenoma, etc.) [7]. It is worth noting that some of these benign lesions, namely IPMN, may have a premalignant nature. Therefore, they often require continuous monitoring to ensure the absence of progression into cancer [8].

Currently, there are no laboratory biomarkers for PDAC that would be sensitive and specific in the early stages of the disease. The only serum biomarker widely used in clinical practice is the carbohydrate antigen 19-9 (CA 19-9). It is characterized by a relatively high sensitivity (up to 73%) and specificity (up to 86%) in patients with locally advanced and metastatic cancer, but it is extremely ineffective for screening due to the low positive predictive value (0.5–0.9%) [9,10]. Moreover, approximately 10% of patients in the PDAC population do not synthesize CA 19-9 at all (due to a negative phenotype for the Lewis antigen), which also limits the use of this biomarker [11].

Recently, a new approach for biomarker analysis in cancer has emerged, namely liquid biopsy. This term refers to a set of minimally invasive techniques for the detection of various tumor derivatives in biological fluids [12]. Targets for liquid biopsy include cell-free tumor DNA (cftDNA), exosomes, microRNA, circulating tumor cells, and others [13]. In the case of PDAC, most studies focused on cftDNA analysis in plasma, which yields promising results for both diagnostic and prognostic purposes [14]. cftDNA is often the biomarker of choice due to the fact that up to 95% of these tumor cells carry activating *KRAS* mutations, mostly in codons 12, 13, and 61 [15]. This feature of PDAC enables the use of relatively cheap targeted approaches for DNA analysis, namely various forms of polymerase chain reaction (PCR), instead of sequencing.

However, the above-mentioned point mutations are not the only frequent genetic alterations related to the *KRAS* gene. It is known that *KRAS* copy number gain (CNG) is detectable in 11–80% (depending on the stage of the disease) of PDACs [16,17]. Hence, cftDNA carrying *KRAS* CNG might be a suitable biomarker for liquid biopsy in patients with this disease. A recent study demonstrated that it was possible to detect this genetic alteration in plasma of several treatment-naïve patients with PDAC, and to our knowledge it is the only available study on the topic in the literature to date [18].

It is worth noting that in general the analysis of point mutations in plasma cftDNA is quite challenging due to its exceptionally low levels (often as low as five copies per 1 mL of plasma) [19]. In the case of CNG analysis, this might be even more detrimental, as small copy number variations (CNV) must be distinguished from the analytical variance of the assay. It is expected that this parameter could be quite high, given the overall scarcity of the cell-free DNA (cfDNA) in plasma.

Our group has demonstrated that bile is superior to plasma in terms of cftDNA/cfDNA content and enables a two-fold increase in cftDNA detection rate [20,21]. It is known that approximately 70% of patients develop biliary obstruction at the time of the initial diagnosis of PC, making the collection of this biomaterial a natural part of the routine treatment [22]. Therefore, for the purposes of liquid biopsy, it might be worth exploring *KRAS* CNG not only in plasma but also in bile.

The aim of this study was to evaluate the diagnostic and prognostic potential of *KRAS* CNG analysis-based liquid biopsy of plasma and bile in patients with PDAC and various other neoplasms of the pancreas.

## 2. Results

### 2.1. Diagnostic Performance of the KRAS CNG Analysis in cfDNA

Analysis of *KRAS* CNG was successfully performed in all the samples collected from the 149 study participants. An example of 2D diagrams generated using ddPCR for the developed *KRAS* CNG detection assay is available in Appendix A. Detailed analytical results for each sample are presented in Appendix A. *KRAS* CNG was not detected in any of the control group samples, as well as in most of the samples from the other pancreatic neoplasms (OPN) group. For the latter there was a single case of plasma *KRAS* CNG positivity in a patient with a main duct IPMN. It is worth mentioning that this sample was characterized by a quite low *KRAS* CNG (copy number (CN) of 2.225) and an absence of cftDNA carrying common point mutations in the *KRAS* G12, G13, and Q61 hotspots. On the other hand, in the PDAC group, *KRAS* CNG was detectable in a significantly higher portion of the plasma samples—in 15 out of 94 cases.

ROC curves illustrating the discrimination potential of *KRAS* CNG analysis in plasma cfDNA are presented in Figure 1. At the predefined *KRAS* CN cutoff value of 2.175, the overall sensitivity and specificity were 16% and 100% (PDAC vs. control)/16% and 94% (PDAC vs. OPN) with areas under the ROC curves (AUC-ROC) of 0.580 and 0.554, respectively. These AUC-ROC values were significantly lower than those obtained for the analysis of cftDNA carrying *KRAS* mutations in the G12, G13, and Q61 hotspots (0.805 and 0.794, respectively). However, a combination of these parameters (*KRAS* CNG and point mutations analysis) based on the “and/or” principle allowed for even better results: AUC-ROC of 0.824 with a sensitivity of 65% and a specificity of 100% (PDAC vs. control); AUC-ROC of 0.801 with a sensitivity of 65% and a specificity of 88% (PDAC vs. OPN).

In the *KRAS* CNG-positive cases (PDAC group), CN values for this gene in plasma cfDNA varied significantly: 2.424 [2.260; 2.628] with a maximum value as high as 2.860 (Figure 2a). These positive cases did not differ from the rest of the PDAC group in terms of the detection rate of *KRAS* point mutations in the G12, G13, and Q61 hotspots (*p* = 0.389, Figure 2d) and levels of cftDNA carrying these mutations (*p* = 0.215, Figure 2c). There was a trend towards higher mutant allele fractions in *KRAS* CNG-positive plasma samples, but the difference did not reach statistical significance (*p* = 0.062, Figure 2b). No correlation between the *KRAS* CN and qualitative parameters of the cftDNA carrying *KRAS* point mutations in the G12, G13, and Q61 hotspots was found either (*p* > 0.05, Appendix A).

Comparison of the *KRAS* CNG analysis results in paired samples of plasma and bile in the PDAC group is presented in Figure 3. Bile exhibited an equal *KRAS* CNG detection rate (4/20 vs. 1/20, *p* = 0.375), with a maximum CN value as high as 3.166. The overall detection rate in the total cohort for bile was equal to that for plasma (20% vs. 16%, *p* = 0.197) as well. It is worth noting that there was a single case of the detection discrepancy: a plasma sample was positive for *KRAS* CNG, but the paired bile sample was not (the opposite situations are not surprising, as bile is expected to be a superior source of cftDNA [20,21]).

### 2.2. Relation of KRAS CNG in cfDNA to Clinical and Demographic Parameters

Results of *KRAS* CNG analysis in plasma cfDNA were not associated with such parameters as age, sex, serum CA 19-9 levels, presence of distant metastases, and tumor’s localization, size, contact with arteries/veins, and invasion into bile ducts (*p* > 0.05, Appendix A). Due to the low sample size for the *KRAS* CNG-positive cases in bile, the assessment of its association with the above-mentioned clinical and demographic parameters was not possible.

In this study, the follow-up period after the biomaterial collection was up to 2 years. Results of the survival analysis for all studied *KRAS*-related cftDNA parameters are presented in Table 1. It appeared that the probability of survival was slightly lower in patients with detectable *KRAS* CNG in plasma compared to those in whom it was undetectable (Figure 4a). However, these differences did not reach statistical significance (*p* = 0.080). On the contrary, the detection of point mutations in the G12, G13, and Q61 hotspots of the *KRAS* gene yielded significant results (*p* = 0.026, Figure 4b), whereas the combined parameter (*KRAS* CNG + point mutations, based on the “and/or” principle) allowed for an even better discrimination (*p* = 0.009, Figure 4c).

To further strengthen the prognostic potential of the studied *KRAS*-related cftDNA parameters, ROC analysis was carried out to establish certain cutoff values for the 2-year follow-up period (CN of 2.287 and cftDNA level of 4.735 copies/mL [21]). At this threshold *KRAS* CNG became significantly associated with poor survival (*p* = 0.001, Figure 4d). The same margin of improvement was observed for the *KRAS* point mutations analysis (*p* = 0.007, Figure 4e). Finally, at the above-mentioned cutoff values, the combined parameter (*KRAS* CNG + point mutations, based on the “and/or” principle) yielded results significantly surpassing the previous models (*p* < 0.001, Figure 4f). In the studied cohort, multivariate Cox regression revealed that these factors were predicting poor survival in a fashion independent from other clinical and demographic parameters.

It is worth noting that the simultaneous presence of *KRAS* CNG and cftDNA carrying *KRAS* point mutations in the G12, G13, and Q61 hotspots in plasma was not associated with the prognosis (*p* > 0.05). The same was true for the analysis of these parameters in bile (*p* > 0.05).

## 3. Discussion

CNVs are the most common genetic structural changes, accounting for approximately 12% of the human genome [23]. Initially, when these genetic alterations were discovered, it was believed that they were not functionally significant but merely reflected the evolutionary path of a given species [24]. However, recent advances in the development of molecular genetic analysis techniques have opened new opportunities to investigate and partially uncover the role of CNVs in the pathogenesis of many diseases, including cancer [25]. In the case of PDAC, there is increasing evidence of a connection between CNVs and the functional and clinical characteristics of the tumor, such as growth rate, resistance to chemotherapy, potential for metastasis, etc. [16]. A recent study has shown that *KRAS* CNG is associated with the molecular subtype of the neoplasm: “basal-like” cells, more prone to metastasis, usually demonstrate the *KRAS* CN value of 3 or even 4, while cells with a “classical” phenotype are characterized by an unaltered CN of this gene [17]. Therefore, CNV analysis, and *KRAS* CNG detection in particular, might become a useful tool for both diagnostic and prognostic purposes in PDAC.

It is known that tumor tissue, even in the primary focus, is characterized by significant heterogeneity [26]. Thus, it is difficult to assess the genetic landscape of the tumor by examining biopsy material, not to mention the serious risks associated with the procedure [27]. However, liquid biopsy is largely devoid of procedure-associated complications due to its minimally invasive nature and enables the detection of cftDNA, which can reflect genetic alterations across all neoplastic cells. According to our data, *KRAS* CNG appears to be a rare event in plasma cfDNA—it was detected only in 16% of samples. These results confirm the findings of the only study published to date on the topic of the *KRAS* CNG analysis in plasma cfDNA (a detection rate of 12.7% in PDAC patients) [18]. The cftDNA content in plasma is quite low (rarely higher than 1% of total cfDNA [21]), and it is not expected to enable the detection of *KRAS* CNG at the same frequency as it would in a tumor tissue sample. However, it is worth noting that even the exact detection rate of this genetic alteration in the PDAC tissue is not known, as data on the topic are limited and vary significantly (11–80%) [16,17]. As for its prevalence in other pancreatic neoplasms included in the OPN group, the data are even more scarce, and it is impossible to provide an estimate [28,29]. In this study, we were able to detect *KRAS* CNG only once in the OPN group, in a plasma sample from a patient with an IPMN. The clinical significance of this discovery requires further investigation with a larger sample size and robust data regarding the progression of the disease. Based on our findings, in combination with the analysis of *KRAS* mutations in the G12, G13, and Q61 hotspots, *KRAS* CNG allows quite efficient discrimination for PDAC (AUC-ROC of 0.824 and 0.801 compared to healthy individuals and patients with OPN, respectively). However, the diagnostic performance of the isolated *KRAS* CNG analysis in plasma appears to be suboptimal.

To our knowledge, this is the first study to report the detectability of *KRAS* CNG in bile samples collected from PDAC patients. According to the literature, bile is superior to plasma in terms of both cftDNA detection rates and its absolute levels (at least for cftDNA carrying point mutations in various hotspots) [21,30,31]. The same was not true for the *KRAS* CNG analysis in cfDNA, as it was demonstrated in the present study. In the paired samples, the detection rate of *KRAS* CNG in bile was slightly higher than in plasma, although the difference did not reach statistical significance. In addition, the highest *KRAS* CN value (3.17) in the total cohort was detected in a bile sample. Particular attention should be given to the results obtained from the paired samples of Patient #5 (Figure 3). It was the only case when an alteration (point mutation or CNG) was detected in plasma but not in paired bile. Interestingly, in this pair of samples, plasma was superior to bile in terms of cftDNA MAF (point mutations in the *KRAS* G12/G13 hotspots), which was a rare phenomenon in our cohort. Moreover, the bile sample from this pair was the only bile sample overall to be positive for cftDNA carrying point mutations in the *KRAS* Q61 hotspot. It is worth mentioning that this patient did not have any known metastases, but the invasion of the tumor into bile ducts and blood vessels was confirmed. This might resemble an intriguing case of a tumor containing two genetically and anatomically distinct cell populations, one of which is carrying a *KRAS* point mutation in the Q61 hotspot and secreting cftDNA into bile, while the other has *KRAS* CNG and point mutations in the *KRAS* G12/G13 hotspots and is secreting cftDNA into blood. The observation described above highlights the incredible utility of liquid biopsy as a tool for analyzing genetic heterogeneity of a tumor.

We were unable to identify any statistically significant associations between *KRAS* CNG in cfDNA and most clinical and demographic characteristics of the study participants. This might be explained by the low detection rate of this genetic alteration and, therefore, by the low sample size of positive cases for the regression analysis. According to our previous report on the same cohort, cftDNA carrying *KRAS* point mutations in the G12, G13, and Q61 hotspots was significantly associated with such parameters as serum CA19-9 levels, presence of distant metastases, and the size of the tumor [21]. There is plenty of supporting data available in the literature on the relationship between these clinical parameters in PDAC patients and cftDNA identified based on the presence of various point mutations [32,33,34], whereas data for *KRAS* CNG in cfDNA on the topic are quite limited [18].

In this study, *KRAS* CNG in plasma cfDNA appeared to be significantly associated with the overall survival of PDAC patients. Moreover, when combined with the analysis of *KRAS* point mutations in the G12, G13, and Q61 hotspots, this genetic alteration allowed for the prediction of poor survival even better. This is in line with the results of a recent pilot study conducted by S. Mohan et al. (an HR of 3.47 [95% CI: 1.19; 10.17], *p* < 0.05 for *KRAS* CNG) [18]. However, we cannot confirm the other finding of the above-mentioned pilot study regarding the worst prognosis in patients with the simultaneous presence of *KRAS* CNG and point mutations. In our cohort, these patients did not differ from the rest in terms of survival, which might be explained by the difference in the inclusion criteria (in the above-mentioned report study participants were predominantly with locally advanced and metastatic disease), sample size (55 compared to 94 in our study), and *KRAS* CNG detection methodology (NGS compared to our ddPCR assay).

Our study had certain limitations. To begin, the control group consisted of volunteers who were younger compared to other groups (the purpose of this group was mainly to ensure the absence of the false-positive signal). However, it is worth noting that *KRAS* CNG is not commonly associated with the clonal hematopoiesis of indeterminate potential (CHIP), a common age-related condition where blood stem cells acquire mutations and expand [35]. CHIP is believed to be the main cause of the presence of certain mutations in cfDNA of seemingly cancer-free individuals [35]. Moreover, it was reported that CHIP has a low impact on the analysis of point mutations in the *KRAS* gene [36]. Therefore, it was not expected that the absence of a perfect age match in the control group could have severely impacted our results. In clinical settings, the analysis would have been primarily performed for differential diagnosis of patients with certain pancreatic diseases, which is why the OPN group was included in our study protocol. Besides this, we were unable to assess the *KRAS* CNG status of matched tumor tissue, which may be useful to verify the absence of false CNG calls (however, this might be compensated with the absence of false-positive results in the control group and the robust determination of the CNG call cutoff value in experiments with wild-type DNA). Next, due to the generally low sample size and the exploratory nature of this study, cutoff values established using the ROC analysis were not cross-validated. Bile samples were available only for a limited number of patients, which, given the low *KRAS* CNG detection rate in this cohort, could have influenced the significance of statistical analyses for data corresponding to this biomaterial. It is worth noting that all comparisons and analyses involving plasma *KRAS* CNG data in this study had sufficient statistical power (α = 0.05 and β = 0.80). Furthermore, the *KRAS* amplicon size in our ddPCR assay was 82 bp, which could negatively influence the detection rate, as, according to Truty et al., ~30% of CNVs in PDAC do not encompass the full length of the gene [37]. Finally, due to limited availability of clinical data for some patients, we were unable to analyze the progression-free survival in the cohort.

## 4. Materials and Methods

### 4.1. General Information

The protocol of this study was reviewed and approved by the Local Ethics Committee of the institution (protocol No. 12/21, 13 December 2021). This study was carried out according to the tenets of the Declaration of Helsinki and its later amendments. Patient recruitment was conducted in a private hospital; the follow-up period after the enrollment was up to 2 years. All patients provided signed informed consent forms. This study included 149 individuals, of whom 94 patients had PDAC (20 had obstructive jaundice at the time of the inclusion and donated bile), 17 patients had various other pancreatic neoplasms (OPN group), and 69 participants were volunteers (control group) without any known oncological diseases. Since all the patients from the above-mentioned cohort took part in our previous study, a detailed description of the methodology regarding morphological verification of the diagnosis, imaging techniques, and routine laboratory biomarker analysis can be found in the corresponding publication [21]. The relevant demographic and clinical characteristics of the patients in each group are presented in Table 2, whereas detailed depersonalized data for each study participant are available in Appendix A.

### 4.2. Biomaterial Collection and Processing

Study participants donated 10–12 mL of peripheral venous blood (collected into EDTA tubes). Biomaterial was obtained before any invasive diagnostic procedures, surgical intervention, and chemotherapy. Then, plasma was immediately separated by centrifugation (3000× *g*, 10 min). Resulting plasma samples were centrifuged again (3000× *g*, 10 min) to ensure the removal of any remaining particles. Biomaterial was stored in fresh low-DNA-binding tubes at a temperature of –80 °C.

PDAC patients with obstructive jaundice underwent external biliary drainage or stenting of the common bile duct as a part of their routine treatment (*n* = 20). In these cases, bile samples of approximately 15 mL were collected and immediately frozen at a temperature of –80 °C.

Upon defrosting, both plasma and bile samples were thoroughly mixed by pulse-vortexing. Then, bile samples underwent centrifugation at conditions described above for peripheral venous blood. cfDNA was isolated from 5 mL of plasma/bile supernatants using the QIAamp Circulating Nucleic Acid Kit (Qiagen GmbH, Hilden, Germany) with the addition of carrier RNA according to the instruction manual. However, for bile supernatant, the lysis stage was extended for an additional 30 min. DNA elution volumes were set to 50 µL.

### 4.3. Cell-Free DNA Analysis

Digital droplet PCR (ddPCR) was the analytical method of choice. DNA amplification was carried out using the Veriti Thermal Cycler (ThermoFisher Scientific, Inc., Waltham, MA, USA). Droplet generation and reading were performed using the QX200 AutoDG ddPCR System (Bio-Rad Laboratories, Inc., Hercules, CA, USA).

*KRAS* CNV analysis was carried out using a self-designed assay (Table 3) paired with the ddPCR Supermix for Probes (Bio-Rad Laboratories, Inc., Hercules, CA, USA). The *EIF2C1* gene located on chromosome 1 was selected as a copy number reference. DNA samples were added into the final ddPCR mixture in a maximum possible volume—9.9 μL. DNA restriction prior to the amplification was not implemented due to the fact that cfDNA is naturally highly fragmented [38]. Optimal annealing/extension temperature for the assay was established in a series of experiments with a gradient of temperatures. The following thermocycling protocol was used: incubation at 95 °C (10 min); 40 cycles of denaturation at 94 °C (30 s) and annealing/extension at 52 °C (1 min); and incubation at 98 °C (10 min). Samples were analyzed in a single measurement. ddPCR was repeated for wells that did not meet the quality criteria (generation of at least 10,000 droplets). Batch variability across multiple reactions was controlled based on the assessment of a CN deviation (no more than 10%) of a control DNA sample purified from the Raji cell line (Evrogen, JSC, Moscow, Russia) and restricted using the FastDigest HindIII enzyme (ThermoFisher Scientific, Inc., Waltham, MA, USA).

The results of *KRAS* CNV analysis in the cell-free DNA were presented using the following formula:CN*_KRAS_* = (C*_KRAS_*/C*_EIF2C1_*) × 2,(1)
where C is the DNA level of the corresponding gene expressed in copies per mL of ddPCR reaction mixture and CN is the copy number. The analytical variance of the developed assay was determined in a series of experiments (*n* = 48; run in duplicates) with *KRAS* wild-type DNA purified from the Raji cell line (Evrogen, JSC, Moscow, Russia) and restricted using the FastDigest HindIII enzyme (ThermoFisher Scientific, Inc., Waltham, MA, USA) at various concentrations expected in cell-free DNA samples of plasma and bile (range: 0.1–1.0 ng/μL), and the cutoff for the *KRAS* CNG call was set to CN*_KRAS_* ≥ 2.175 based on the highest observed value.

Data regarding the levels of cftDNA carrying *KRAS* mutations G12A, G12C, G12D, G12R, G12S, G12V, and G13D and Q61H (183A > C), Q61H (183A > T), Q61K, Q61L, and Q61R in the studied samples were extracted from our previous report [21].

### 4.4. Statistical Analysis

Data analysis and processing were carried out using IBM SPSS Statistics 26.0 Software (IBM Corp., Armonk, NY, USA). The Shapiro–Wilk test was used to determine the data distribution normality. Normal distribution was absent for all studied variables. Thus, non-parametric statistical tests were implemented for further analysis. Numerical variables are presented as medians [quartile 1; quartile 3], unless stated otherwise. Unpaired categorical and numerical variables were compared using Fisher’s exact and Mann–Whitney U tests, respectively. Paired categorical and numerical variables were compared using McNemar’s and Wilcoxon tests, respectively. Receiver operating characteristic (ROC) analysis was used to determine the binary classification potential for the studied variables; cutoff values were determined using the Youden index. Spearman’s rank correlation coefficient (r_S_) was used to assess dependency between the numerical variables; the strength of the relationships was assessed using Chaddock’s scale. Kaplan–Meier plots and the log-rank (Mantel–Cox) test were used to carry out the survival analysis. Cox proportional hazards regression analysis was implemented to determine the hazard ratios (HRs) and the corresponding 95% confidence intervals (CIs). A *p*-value below 0.05 was considered statistically significant.

## 5. Conclusions

CNVs are an important yet underexplored aspect of cancer molecular biology. Our exploratory study has demonstrated that not only are they detectable in cfDNA of plasma and bile, but also they might be relevant to predicting poor survival in PDAC patients. The prognostic potential of *KRAS* CNG in plasma cfDNA, especially in combination with point mutations in this gene, might be useful to guide therapeutic decisions, for example, to identify patients requiring neoadjuvant chemotherapy or those who may benefit from additional imaging prior to treatment. However, the clinical implications of this kind of cftDNA analysis require further investigation. The ddPCR assay for *KRAS* CNG analysis presented in this work might be easily reproduced in any laboratory equipped with a ddPCR instrument, and we hope that it will be useful for future studies on the topic in PDAC and beyond.

## Figures and Tables

**Figure 1 ijms-26-08763-f001:**
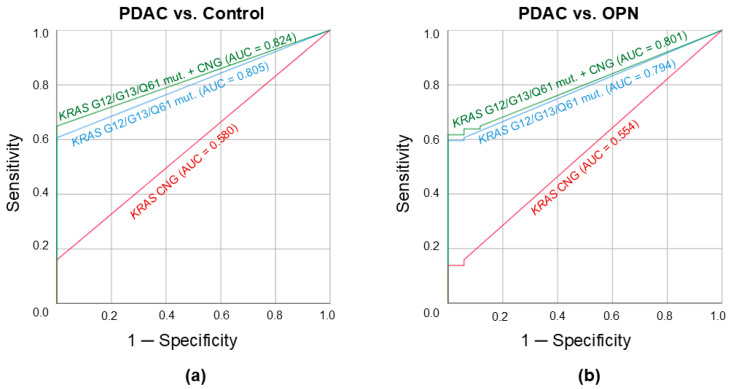
ROC curves for plasma cell-free tumor DNA analysis. (**a**) Discrimination potential for PDAC and healthy control groups. (**b**) Discrimination potential for PDAC and OPN groups. Red curves correspond to *KRAS* CNG; blue curves—various mutations in the *KRAS* G12, G13, and Q61 hotspots; green curves—combined parameter (*KRAS* CNG and various mutations in the *KRAS* G12, G13, and Q61 hotspots). PDAC, pancreatic ductal adenocarcinoma; OPN, other pancreatic neoplasms; CNG, copy number gain; mut., mutations; AUC, area under the ROC curve.

**Figure 2 ijms-26-08763-f002:**
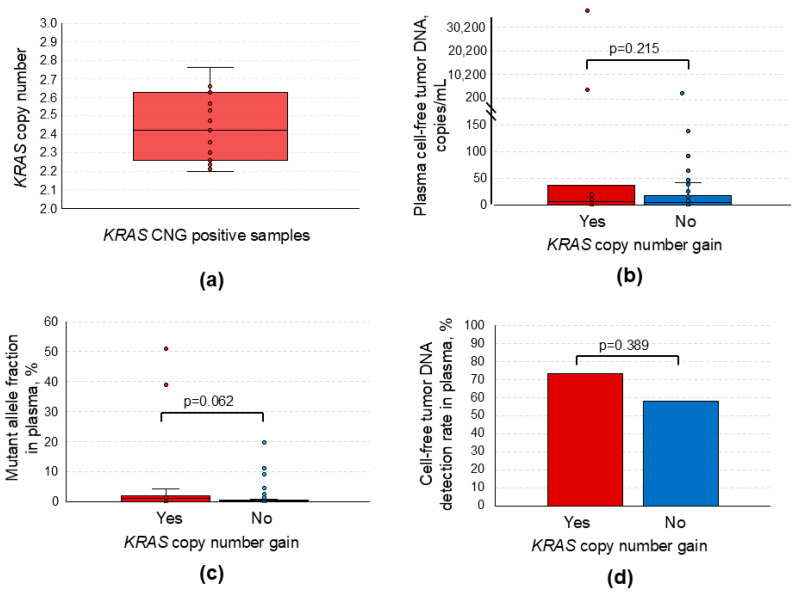
Distribution of *KRAS* copy number variation in the *KRAS* CNG positive samples and its relation to plasma cell-free tumor DNA carrying common *KRAS* mutations in the G12, G13, and Q61 hotspots for the PDAC group. (**a**) Box plot for *KRAS* copy number distribution. (**b**) Box plot for cftDNA levels presented as copies per 1 mL of plasma. (**c**) Box plot for cftDNA presented as the mutant allele fraction. (**d**) Column chart for data presented as the cftDNA detection rate. Colored dots on the box plots represent each data point for the analyzed variables. The *KRAS* CNG call threshold was set to 2.175 based on the results of the preclinical assay validation. PDAC, pancreatic ductal adenocarcinoma; CNG, copy number gain; cftDNA, cell-free tumor DNA.

**Figure 3 ijms-26-08763-f003:**
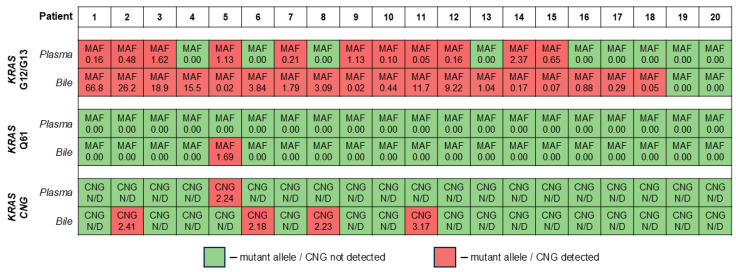
Comparison of cell-free tumor DNA analysis results in paired samples of plasma and bile in the PDAC group. Data are presented for both cell-free tumor DNA carrying common *KRAS* mutations in the G12, G13, and Q61 hotspots (mutant allele fractions) and *KRAS* CNG (copy number values). Comparison of data in paired samples (plasma and bile) using statistical tests for *KRAS* CNG was not performed due to low sample size for CNG-positive cases. CNG, copy number gain; MAF, mutant allele fraction.

**Figure 4 ijms-26-08763-f004:**
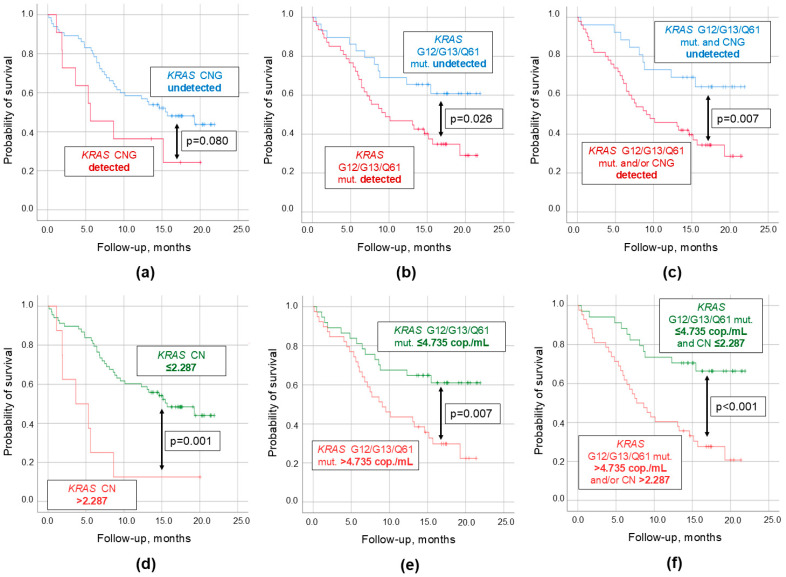
Caplan–Meier survival curves for plasma cell-free tumor DNA analysis in patients with pancreatic ductal adenocarcinoma. (**a**) Qualitative threshold for *KRAS* CNG. (**b**) Qualitative threshold for cell-free tumor DNA carrying common *KRAS* mutations in the G12, G13, and Q61 hotspots. (**c**) Qualitative threshold for a combined parameter (CNG and/or point mutations). (**d**) Quantitative threshold for *KRAS* CNG (>2.287). (**e**) Quantitative threshold for cell-free tumor DNA carrying common *KRAS* mutations in the G12, G13, and Q61 hotspots (>4.735 cop./mL). (**f**) Quantitative threshold for a combined parameter (CNG and/or point mutations at above-mentioned thresholds). Censored observations were marked with a cross. Threshold values were established using ROC analysis for the 2-year survival. CNG, copy number gain; CN, copy number; cop., copies; mut., mutations.

**Table 1 ijms-26-08763-t001:** Associations of KRAS mutation analysis in plasma cell-free DNA with the survival in PDAC patients.

Parameters	Survival, Months	HR (95% CI)	*p*-Values
Detectable *KRAS* CNG Undetectable *KRAS* CNG	9.3 ± 2.2 14.1 ± 1.0	1.92 (0.91–4.27)	0.080
Detectable *KRAS* point mutations Undetectable *KRAS* point mutations	11.8 ± 1.1 16.1 ± 1.5	2.15 (1.08–4.29)	0.026
Detectable *KRAS* CNG and/or point mutations Undetectable *KRAS* CNG and point mutations	11.6 ± 1.1 17.0 ± 1.4	2.58 (1.23–5.41)	0.009
*KRAS* CNG > 2.287 *KRAS* CNG ≤ 2.287	6.0 ± 2.0 14.3 ± 1.0	3.54 (1.55–8.12)	0.001
*KRAS* point mutations > 4.735 copies/mL *KRAS* point mutations ≤ 4.735 copies/mL	11.6 ± 1.1 17.0 ± 1.4	2.35 (1.23–4.48)	0.007
*KRAS* CNG and/or point mutations > thresholds ^1^ *KRAS* CNG and point mutations ≤ thresholds ^1^	10.5 ± 1.2 17.1 ± 1.3	3.23 (1.62–6.44)	<0.001

PDAC, pancreatic ductal adenocarcinoma; CNG, copy number gain; HR, hazard ratio; CI, confidence interval; ^1^ threshold for KRAS CNG was 2.287, whereas for KRAS point mutations it was 4.735 copies/mL. Threshold values were established using ROC analysis for the 2-year survival.

**Table 2 ijms-26-08763-t002:** The demographic and clinical characteristics of study participants.

Parameters	PDAC Group (*n* = 94)	OPN Group (*n* = 17)	Control Group (*n* = 69)
Age, years ^1^	65 (41–88)	58 (43–66)	40 (19–71)
Sex, *n* (%):			
- Male	46/94	4/17	33/69
- Female	48/94	13/17	36/69
OPN types:			
- mdIPMN, *n*	N/A	5/17	N/A
- multIPMN, *n*	N/A	1/17	N/A
- bdIPMN, *n*	N/A	7/17	N/A
- adenoma, *n*	N/A	1/17	N/A
- serous cystadenoma, *n*	N/A	1/17	N/A
- SPPN, *n*	N/A	1/17	N/A
- NET, *n*	N/A	1/17	N/A
Tumor localization in pancreas:			
- head, *n*	44/87 ^2^	N/A	N/A
- body, *n*	6/87 ^2^	N/A	N/A
- tail, *n*	11/87 ^2^	N/A	N/A
- head + body, *n*	10/87 ^2^	N/A	N/A
- body + tail, *n*	15/87 ^2^	N/A	N/A
- head + body + tail, *n*	1/87 ^2^	N/A	N/A
Tumor size:			
- >4 cm, *n*	39/87 ^2^	N/A	N/A
- 2–4 cm, *n*	41/87 ^2^	N/A	N/A
- <2 cm, *n*	7/87 ^2^	N/A	N/A
Contact with arteries/veins, *n*	78/87 ^2^	N/A	N/A
Bile ducts invasion, *n*	39/87 ^2^	N/A	N/A
Distant metastases, *n*	43/87 ^2^	N/A	N/A
CA 19-9, U/mL ^3^	36.5 [0.9; 883.3]	3.9 [0.8; 7.9]	N/A

PDAC, pancreatic ductal adenocarcinoma; OPN, other pancreatic neoplasms; N/A, not available/not applicable; mdIPMN, main duct intraductal papillary mucinous neoplasm; multIPMN, multifocal intraductal papillary mucinous neoplasm; bdIPMN, branch duct intraductal papillary mucinous neoplasm; SPPN, solid pseudopapillary neoplasm; NET, neuroendocrine tumor; ^1^ data presented as mean (range); ^2^ data not available for 7/94 patients; ^3^ data presented as median [quartile 1; quartile 3]. This table was partially reproduced from our previous publication [21].

**Table 3 ijms-26-08763-t003:** Primers and probes used for the *KRAS* CNV detection.

Gene.	Amplicon Size	Oligonucleotide	Sequence	Concentration ^1^
*KRAS*	82 bp	Forward primer	GTA ATT TAC TGG GAA AGC	0.9 μM
		Reverse primer	CAG TCT GAT GTC TGT TTA	0.9 μM
		Probe	FAM-AGC TCA TAA TCT CAA ACT TCT TGC ACA-BHQ1	0.25 μM
*EIF2C1*	81 bp	Forward primer	GTT CGG CTT TCA CCA GTC T	0.9 μM
		Reverse primer	CTC CAT AGC TCT CCC CAC TC	0.9 μM
		Probe	HEX-CGC CCT GCC ATG TGG AAG AT-BHQ1	0.25 μM

^1^ Concentration in the final digital droplet PCR mixture. FAM, 6-Carboxyfluorescein; HEX, Hexachlorofluorescein; BHQ1, Black Hole Quencher 1.

## Data Availability

All data generated in the present study are available in Appendix A.

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
