# Peer review of "KRAS Copy Number Gain in Cell-Free DNA Analysis-Based Liquid Biopsy of Plasma and Bile in Patients with Various Pancreatic Neoplasms"

_ijms, 2025, doi:10.3390/ijms26188763_

Round 1
Reviewer 1 Report
Comments and Suggestions for Authors
In this study, the authors investigate the diagnostic and prognostic potential of KRAS copy number gain (CNG) in plasma and bile cfDNA from pancreatic ductal adenocarcinoma (PDAC) patients using ddPCR. Below are a number of issues that the authors shall address or revise:
1. Figure 1, ROC curves should be checked. ROC curves are curves and not broken lines.
2. The calculation of the KRAS CN cutoff value should be described in detail.
3. Table 2, the control group (age 20±2 years) is significantly younger than the PDAC group (65±12 years). Age-matching is critical, as cfDNA can vary with age.
4. The findings of cfDNA were not validated with tumor tissues or an external dataset. The authors should give more evidence to support their results.
5. The processing of human blood and bile should be described.
Author Response
Esteemed Reviewer,
We are grateful for the time and effort dedicated to providing valuable comments on our manuscript. We have done our best to revise the manuscript and include all the suggestions provided. The changes are highlighted in the revised manuscript using MS Word tools.
Below you may find a point-by-point response to received comments.
Comment 1: Figure 1, ROC curves should be checked. ROC curves are curves and not broken lines.
Response: Thank you for this comment. We have checked Figure 1 and consulted with a specialist in statistics. In our case ROC-curves do not have many points due to a specific reason: there were no false-positives (compared to the control group) and almost no false-positives (compared to the OPN group; although we cannot claim that in the case of the OPN group these were false-positives, as these values were above the predefined threshold). In Figure 1, ROC-curves were meant to reflect the classification potential of the test overall and to validate our predefined threshold. If our goal was to determine the threshold value using this figure, there would indeed be many points on the curves. The fact that in Figure 1a there is only a single point on the ROC-curve highlights that our selection of threshold value was accurate.
Comment 2: The calculation of the KRAS CN cutoff value should be described in detail.
Response: Thank you for pointing this out. We agree that we rushed the description of this aspect of methodology. Corresponding information has been added to the materials and methods section:
“The analytical variance of the developed assay was determined in a series of experiments (n=48; ran in duplicates) with KRAS wild-type DNA purified from Raji cell line (Evrogen, JSC, Moscow, Russia) and restricted using the FastDigest HindIII enzyme (ThermoFisher Scientific, Inc., Waltham, Massachusetts, USA) at various concentrations expected in cell-free DNA samples of plasma and bile (range: 0.1–1.0 ng/μL), and the cutoff for KRAS CNG call was set to CNKRAS ≥ 2.175 based on the highest observed value.”
Comment 3: Table 2, the control group (age 20±2 years) is significantly younger than the PDAC group (65±12 years). Age-matching is critical, as cfDNA can vary with age.
Response: We are grateful for this comment. We have expanded our control group (+31 plasma samples from volunteers without oncological diseases aged 50+ years). Fortunately, our study protocol did not cap us on the size of this group, so we were able to enroll patients from the hospital admissions department. These samples did not exhibit KRAS copy number gain. All relevant information has been added to the corresponding sections of the manuscript. Initially, we did consider the age discrepancy an issue, as Clonal Hematopoiesis of Indeterminate Potential (CHIP), a common age-related condition where blood stem cells acquire mutations and expand. It is known that this condition is the most likely cause of the presence of certain mutations in cell-free DNA of seemingly cancer-free patients, yet it has a low impact on the analysis of point mutations in the KRAS gene (doi:10.3390/DIAGNOSTICS12081956). Moreover, KRAS CNG is not associated with CHIP. We included the age-matched other pancreatic neoplasms group to the study to provide a “true control” group, as in real clinical setting it is most likely that differential diagnosis will take place in cases with suspected pancreatic neoplasms – not in an asymptomatic population. Anyway, we have completely rewritten the study limitations section to clearly highlight this issue:
“Our study had certain limitations. To begin, the control group consisted of volunteers who were younger compared to other groups (the purpose of this group was mainly to ensure the absence of the false-positive signal). However, it is worth noting that KRAS CNG is not commonly associated with the clonal hematopoiesis of indeterminate potential (CHIP), a common age-related condition where blood stem cells acquire mutations and expand [35]. CHIP is believed to be the main cause of the presence of certain mutations in cfDNA of seemingly cancer-free individuals [35]. Moreover, it was reported that CHIP has a low impact on the analysis of point mutations in the KRAS gene [36]. Therefore, it was not expected that the absence of a perfect age-match in the control group could have severely impacted our results. In clinical settings, the analysis would have been primarily performed for differential diagnosis of patients with certain pancreatic diseases, which is why the OPN group was included in our study protocol. Besides this, we were unable to assess the KRAS CNG status of matched tumor tissue which may be useful to verify the absence of false CNG calls (however, this might be compensated with the absence of false-positive results in the control group and a robust determination of the CNG call cutoff value in experiments with wild-type DNA). Next, due to generally low sample size and the exploratory nature of the study cutoff values established using the ROC-analysis were not cross-validated. Bile samples were available only for a limited number of patients, which, given the low KRAS CNG detection rate in this cohort, could have influenced the significance of statistical analyses for data corresponding to this biomaterial. It is worth noting that all comparisons and analyses involving plasma KRAS CNG data in the study had sufficient statistical power (α=0.05 and β=0.80). Furthermore, the KRAS amplicon size in our ddPCR assay was 82 bp which could negatively influence the detection rate, as, according to Truty et al., ~30% of CNVs in PDAC do not encompass the full length of the gene [37]. Finally, due to limited availability of clinical data for some patients, we were unable to analyze the progression-free survival in the cohort.”
Comment 4: The findings of cfDNA were not validated with tumor tissues or an external dataset. The authors should give more evidence to support their results.
Response: We are grateful for this suggestion. Unfortunately, we were unable to obtain tumor tissue from the study participants. This study was conducted in a private hospital with strict policies regarding the material collected during the paid medical procedures (biopsy, resection). Moreover, some individuals withdraw their tumor material for evaluation in other laboratories. In the OPN group none of the patients had undergone surgical intervention, thus it was not possible to obtain this biomaterial as well. In recent years, liquid biopsy has emerged as an alternative to invasive tumor tissue testing, which is highlighted by the fact that according to the NCCN guidelines liquid biopsy testing is recommended where tissue testing is unavailable. Obviously, we have evaluated a less studied genetic alteration which is not included in these guidelines, and additional validation could have been useful, we are unable to do it for this cohort. We believe that the absence of false-positive results in the control group with a robust determination of the CNG call cutoff value in experiments with wild-type DNA could to at least some extent compensate for this drawback (as well as the theoretical basis regarding the fact that these alterations are not associated with CHIP). Please refer to the updated “study limitations" section above, where we clearly highlight this issue. Validation in an external cohort at the moment is also unachievable. It is most relevant for the prognostic aspect of our study. However, this would require enrolling 90+ PDAC patients (given the discovered rarity of KRAS CNG in plasma) with longitudinal monitoring (as it was in the present manuscript). Such external cohort is unavailable for us. We could not find it in any of the regional biobanks as well. Our study had an exploratory nature. It is meant to demonstrate that KRAS CNG, which is often forgotten when speaking of KRAS mutations, is detectable in cell-free DNA of plasma and bile, and that it might be a promising biomarker in PDAC patients. As is stated in the conclusions section, this biomarker requires further validation. Yet, we hope that our study will bring some attention of the scientific community to this topic and facilitate future research.
Comment 5: The processing of human blood and bile should be described.
Response: Thank you for pointing this out. This information was added to the materials and methods section. See below:
“Study participants donated 10–12 mL of peripheral venous blood (collected into EDTA tubes). Biomaterial was obtained before any invasive diagnostic procedures, surgical intervention, and chemotherapy. Then plasma was immediately separated by centrifugation (3000 g, 10 min). Resulting plasma samples were centrifuged again (3000 g, 10 min) to ensure the removal of any remaining particles. Biomaterial was stored in fresh low DNA binding tubes at a temperature of –80 °C.
PDAC patients with obstructive jaundice underwent external biliary drainage or stenting of the common bile duct as a part of their routine treatment (n=20). In these cases, bile samples of approximately 15 mL were collected and immediately frozen at a temperature of –80 °C.
Upon defrosting, both plasma and bile samples were thoroughly mixed by pulse-vortexing. Then, bile samples underwent centrifugation at conditions described above for peripheral venous blood. cfDNA was isolated from 5 mL of plasma / bile supernatants using QIAamp Circulating Nucleic Acid Kit (Qiagen GmbH, Hilden, Germany) with the addition of carrier RNA according to the instruction manual. However, for bile supernatant the lysis stage was extended for additional 30 minutes. DNA elution volumes were set to 50 µL.”
Reviewer 2 Report
Comments and Suggestions for Authors
This manuscript explores KRAS copy number gain (CNG) in cfDNA from plasma and bile as a diagnostic and prognostic marker for pancreatic cancer. The topic is highly relevant, as liquid biopsy approaches are rapidly advancing and the need for better biomarkers in PDAC remains high. It is well structured and clearly written, however there are certain methodological flaws and issues with interpretation and presentation of the data that need to be addressed before considering it for publication in IJMS.
Please find my concerns below.
Major concerns:
- One of my major concerns is the cohort design and controls. The control group is significantly younger than the PDAC group, which introduces potential bias since cfDNA profiles are age-related. In addition only 20 bile samples were available, making bile–plasma comparisons underpowered. I recommend the authors to further discuss these aspects.
- The KRAS CNG assay was developed in-house. I wonder how reliable the method is as it lacks external validation.
- The EIF2C1 gene located on chromosome 1 was selected as a copy number reference. That is, only one reference gene (EIF2C1) was used for normalization. I recommend the authors to provide data for multiple reference samples.
- Authors did not perform any comparison with matched tumor tissue CNV status. Hence it is unclear whether cfDNA results reflect true tumor genetics.
- The clinical significance of the findings is overstated. Hazard ratios for survival are based on very few positive cases, with wide confidence intervals.
- The data do not support the claim that bile cfDNA is superior to plasma as the observed difference is negligible.
Minor concerns:
- Conclusions should be framed as exploratory, not definitive. The authors overinterprets their findings.
- To further improve the clarity and flow of the manuscript, significant rearrangement is needed. Results sections are dense with numerical detail. I recommend summarizing key findings in the text and leaving full numbers in tables and/or figures. Some sentences in the Introduction and Discussion are long and should be shortened for readability.
- While emphasizing the novelty of bile analysis, authors should clearly acknowledge the small sample size they used.
- Certain methods sections appear to be reused from prior publications. Despite the citations, the degree of overlap raises concerns about plagiarism. Consider rephrasing to emphasize what is new in this study.
To further consider this manuscript, I recommend a major revision. Specifically, the authors may provide additional assay validation, improve clarity and reduce redundancy in text and reframe conclusions to highlight exploratory nature. If additional assay validation cannot be provided it should be explained and acknowledge the limitations more explicitly.
Author Response
Esteemed Reviewer,
We are grateful for the time and effort dedicated to providing valuable comments on our manuscript. We have done our best to revise the manuscript and include all the suggestions provided. The changes are highlighted in the revised manuscript using MS Word tools.
Below you may find a point-by-point response to received comments.
- Major concerns
Comment 1: One of my major concerns is the cohort design and controls. The control group is significantly younger than the PDAC group, which introduces potential bias since cfDNA profiles are age-related. In addition only 20 bile samples were available, making bile–plasma comparisons underpowered. I recommend the authors to further discuss these aspects.
Response: We are grateful for this comment. We have expanded our control group (+31 plasma samples from volunteers without oncological diseases aged 50+ years). Fortunately, our study protocol did not cap us on the size of this group, so we were able to enroll patients from the hospital admissions department. These samples did not exhibit KRAS copy number gain. All relevant information has been added to the corresponding sections of the manuscript. Initially, we did consider the age discrepancy an issue, as Clonal Hematopoiesis of Indeterminate Potential (CHIP), a common age-related condition where blood stem cells acquire mutations and expand. It is known that this condition is the most likely cause of the presence of certain mutations in cell-free DNA of seemingly cancer-free patients, yet it has a low impact on the analysis of point mutations in the KRAS gene (doi:10.3390/DIAGNOSTICS12081956). Moreover, KRAS CNG is not associated with CHIP. We included the age-matched other pancreatic neoplasms group to the study to provide a “true control” group, as in real clinical setting it is most likely that differential diagnosis will take place in cases with suspected pancreatic neoplasms – not in an asymptomatic population. Anyway, we have completely rewritten the study limitations section to clearly highlight this issue:
“Our study had certain limitations. To begin, the control group consisted of volunteers who were younger compared to other groups (the purpose of this group was mainly to ensure the absence of the false-positive signal). However, it is worth noting that KRAS CNG is not commonly associated with the clonal hematopoiesis of indeterminate potential (CHIP), a common age-related condition where blood stem cells acquire mutations and expand [35]. CHIP is believed to be the main cause of the presence of certain mutations in cfDNA of seemingly cancer-free individuals [35]. Moreover, it was reported that CHIP has a low impact on the analysis of point mutations in the KRAS gene [36]. Therefore, it was not expected that the absence of a perfect age-match in the control group could have severely impacted our results. In clinical settings, the analysis would have been primarily performed for differential diagnosis of patients with certain pancreatic diseases, which is why the OPN group was included in our study protocol. Next, due to generally low sample size and the exploratory nature of the study cutoff values established using the ROC-analysis were not cross-validated.”
Please refer to the response for Comments 5+6 regarding the sample size for bile.
Comments 2+3: The KRAS CNG assay was developed in-house. I wonder how reliable the method is as it lacks external validation. The EIF2C1 gene located on chromosome 1 was selected as a copy number reference. That is, only one reference gene (EIF2C1) was used for normalization. I recommend the authors to provide data for multiple reference samples.
Response: Thank you for this comment. Indeed, our assay was self-designed. We did not have an opportunity to conduct an external validation, as it would require us to find another PDAC cohort (n=100+, given the observed rarity of this alteration). We searched regional biobanks: unfortunately, there is no way for us to obtain biomaterial of these patients in an adequate quantity. However, the design of this assay follows the idea behind other Copy Number Assessment assays for ddPCR (for example, please see PrimePCR™ ddPCR™ Copy Number Assay kits by Bio-Rad, many of which are wet lab validated). These assays are also using a single gene reference (often the EIF2C1 gene). We selected a conservative region of this gene, which is devoid of genetic polymorphism. Chromosome 1 is known to be quite stable, and aneuploidies of this chromosome are exceptionally rare even in cancer. The concept behind CNV analysis is quite simple, we just need to compare a target to a reference. The main challenge here (given the low abundance of tumor DNA in cfDNA) is to optimize the assay so that it would yield a copy number close to 2 in wild-type DNA. In our experience, even a slight discrepancy of amplicon sizes for target/reference significantly increases the deviation of the analysis results. Our assay was first validated using restricted DNA in concentrations resembling that of cfDNA, then it was validated using a control group – there were no false-positives. To more clearly describe this process the materials and methods section was expanded:
“The analytical variance of the developed assay was determined in a series of experiments (n=48; ran in duplicates) with KRAS wild-type DNA purified from Raji cell line (Evrogen, JSC, Moscow, Russia) and restricted using the FastDigest HindIII enzyme (ThermoFisher Scientific, Inc., Waltham, Massachusetts, USA) at various concentrations expected in cell-free DNA samples of plasma and bile (range: 0.1–1.0 ng/μL), and the cutoff for KRAS CNG call was set to CNKRAS ≥ 2.175 based on the highest observed value.”
The addition of a new reference sequence would be quite time-consuming, and it is hard to estimate the benefit it would achieve for the assay. Moreover, it would double the price of the analysis, whereas we chose ddPCR as a cheaper alternative for CNV analysis, compared to NGS. Our assay has a weak point in the KRAS target sequence, but it is impossible to overcome without using sequencing. Please see this statement in the study limitations section:
“Furthermore, the KRAS amplicon size in our in-house ddPCR assay was 82 bp which could negatively influence the detection rate, as, according to Truty et al., ~30% of CNVs in PDAC do not encompass the full length of the gene [37].”
In our study, we have provided all necessary information about our assay so that it could be easily replicated in any laboratory with a dPCR instrument. We hope that our results will facilitate other studies of KRAS CNG in cfDNA, and that our assay will be useful for other researchers, which will eventually yield a true external validation.
Comment 4: Authors did not perform any comparison with matched tumor tissue CNV status. Hence it is unclear whether cfDNA results reflect true tumor genetics.
Response: We are grateful for pointing this out. Unfortunately, we were unable to obtain tumor tissue from the study participants. This study was conducted in a private hospital with strict policies regarding the material collected during the paid medical procedures (biopsy, resection). Moreover, some individuals withdraw their tumor material for evaluation in other laboratories. In the OPN group none of the patients had undergone surgical intervention, thus it was not possible to obtain this biomaterial as well. In recent years, liquid biopsy has emerged as an alternative to invasive tumor tissue testing, which is highlighted by the fact that according to the NCCN guidelines liquid biopsy testing is recommended where tissue testing is unavailable. Obviously, we have evaluated a less studied genetic alteration which is not included in these guidelines, and additional validation could have been useful, we are unable to do it for this cohort. We believe that the absence of false-positive results in the control group with a robust determination of the CNG call cutoff value in experiments with wild-type DNA could to at least some extent compensate for this drawback (as well as the theoretical basis regarding the fact that these alterations are not associated with CHIP). Please refer to the updated “study limitations" section (response to comment 1) above, where we clearly highlight this issue and to an addition to this section below:
“Besides this, we were unable to assess the KRAS CNG status of matched tumor tissue which may be useful to verify the absence of false CNG calls (however, this might be compensated with the absence of false-positive results in the control group and the robust determination of the CNG call cutoff value in experiments with wild-type DNA).”
Comments 5+6: The clinical significance of the findings is overstated. Hazard ratios for survival are based on very few positive cases, with wide confidence intervals. The data do not support the claim that bile cfDNA is superior to plasma as the observed difference is negligible.
Response: Thank you for pointing this out. Indeed, the confidence intervals are quite big, but in statistically significant cases they do not overlap with the point of “statistical” insignificance. Moreover, in most cases they do not come close to the point of “clinical” insignificance (such as 1.01-1.20; obviously, these values are not defined anywhere, but they might be derived from the clinical perspective of the usefulness of such molecular testing). In figures 4b,c,e,f, it is clear that both curves have many points due to abundance of KRAS mutations overall. However, we have to agree with the Reviewer that for Figures 4a,d red curves are based on a small sample size, which is dictated by the rarity of KRAS CNG in plasma (in Figure 4d the difference between curves is quite evident even in low sample size settings). It is worth mentioning that prior to the analysis we have assessed the power of the tests. It appeared that all tests involving KRAS analysis in plasma had a significant beta value (above 0.8 at alpha 0.05), hence the power was sufficient. Evidently, tests involving only KRAS CNG were at the lower end of the spectrum (as it was a rare alteration). Tests involving bile were underpowered (for KRAS CNG). To clarify this issue, we have modified the study limitations section:
“Bile samples were available only for a limited number of patients, which, given the low KRAS CNG detection rate in this cohort, have influenced the significance of statistical analyses for data corresponding to this biomaterial. It is worth noting that all comparisons and analyses involving plasma KRAS CNG data in the study had sufficient statistical power (α=0.05 and β=0.80). Finally, due to limited availability of clinical data for some patients, we were unable to analyze the progression-free survival in the cohort.”
We apologize for the overstatement regarding bile samples. Initially, in the results section we wrote that the p value was above 0.05, but in the abstract and discussion we stated that it was “slightly higher”. In the revised version of the manuscript all instances were corrected.
Results: “Comparison of the KRAS CNG analysis results in paired samples of plasma and bile in the PDAC group is presented in Figure 3. Bile exhibited an equal KRAS CNG detection rate (4/20 vs. 1/20, p=0.375), with a maximum CN value as high as 3.166. The overall detection rate in the total cohort for bile was equal to that for plasma (20% vs. 16%, p=0.197) as well.”
Discussion: “According to the literature, bile is superior to plasma in terms of both cftDNA detection rates and its absolute levels (at least for cftDNA carrying point mutations in various hotspots) [21,30,31]. The same was not true for the KRAS CNG analysis in cfDNA, as it was demonstrated in the present study. In the paired samples, the detection rate of KRAS CNG in bile was slightly higher than in plasma, although the difference did not reach statistical significance.”
Abstract: “Bile exhibited an equal KRAS CNG detection rate compared to plasma (20% vs. 16%).”
Minor concerns:
Comment 1: Conclusions should be framed as exploratory, not definitive. The authors overinterprets their findings.
Response: We appreciate this suggestion. Indeed, our study had an exploratory nature. It was meant to demonstrate that KRAS CNG, which is often forgotten when speaking of KRAS mutations, is detectable in cell-free DNA of plasma and bile, and that it might be a promising biomarker in PDAC patients. As is stated in the conclusions section, this biomarker requires further validation. Yet, we hope that our study will bring some attention of the scientific community to this topic and facilitate future research. The following corrections to the text have been made:
Abstract: “…KRAS CNG in cfDNA might be a promising biomarker for both diagnostic and prognostic purposes in PDAC.”
Conclusions: “…Our exploratory study has demonstrated that not only are they detectable in cfDNA of plasma and bile but also might be relevant to predict poor survival in PDAC patients. The prognostic potential of KRAS CNG in plasma cfDNA, especially in combination with point mutations in this gene, might be useful to guide therapeutic decisions, for example, to identify patients requiring neoadjuvant chemotherapy or those who may benefit from additional imaging prior to treatment. However, the clinical implications of this kind of cftDNA analysis requires further investigation…”
Comment 2: To further improve the clarity and flow of the manuscript, significant rearrangement is needed. Results sections are dense with numerical detail. I recommend summarizing key findings in the text and leaving full numbers in tables and/or figures. Some sentences in the Introduction and Discussion are long and should be shortened for readability.
Response: We appreciate this suggestion. To improve the clarity of the manuscript, we have removed numerical data which duplicates information presented in Figures and Tables. Please see the highlighted corrections to the Results section. Moreover, we have significantly restructured the second part of this section, which was dedicated to survival analysis. We have added a new table (Table 1) to it, which contains all relevant numerical data (survival, HR, CI, p-values). Additionally, we have shortened a few sentences in both introduction and discussion sections. A few typos were corrected along the way.
Comment 3: While emphasizing the novelty of bile analysis, authors should clearly acknowledge the small sample size they used.
Response: We agree that it is crucial to highlight this limitation. Please see the improved study limitations section above.
Comment 4: Certain methods sections appear to be reused from prior publications. Despite the citations, the degree of overlap raises concerns about plagiarism. Consider rephrasing to emphasize what is new in this study.
Response: Thank you for pointing this out. We were contacted by the Editor and asked to diminish the overlap in the description of the methodology between the two publications. We have introduced corrections to the text accordingly. This manuscript differs significantly from the previous as it is dedicated to KRAS CNG, whereas the previous one was dedicated only to the analysis of point mutations. In the second study biomaterials from a few patients from the previous cohort were unavailable. Thus, in this manuscript we had to recalculate certain data regarding the cftDNA. So, the ROC-curves and survival plots for the analysis of point mutations had to be redrawn, we could not just cite them. Additionally, it was essential to study the relationship between KRAS CNG and the abundance of point mutations in this gene. This was impossible without the data obtained from previous work. To clarify the amount of data extracted from the previous publication we have added the following sentence:
“Data regarding the levels of cftDNA carrying KRAS mutations G12A, G12C, G12D, G12R, G12S, G12V, and G13D, and Q61H (183A>C), Q61H (183A>T), Q61K, Q61L, and Q61R in the studied samples were extracted from our previous report [21].”
We assure the Reviewer that there was no self-plagiarism in the manuscript, as the overlap is minor, all data regarding KRAS CNG is original and was not published elsewhere. The description of ddPCR in the materials and methods might be similar, but the thermocycling conditions, quality control criteria, primers and probes, concentrations of the reagents are different. It was impossible for us to completely rephrase it, as the instrument is the same in both studies. Table 2 was partially reproduced from the previous publication (which is clearly stated in the text). Again, it was impossible for us to just cite the previous work, as the cohorts are slightly differing.
Round 2
Reviewer 1 Report
Comments and Suggestions for Authors
I am satisfied with the author’s responses to my issues raised in my initial review. The revised manuscript is easier to follow based on feedback from the reviewers. I recommend that the revised paper be accepted.